# The Expression of Connexin 26 Regulates the Radiosensitivity of Hepatocellular Carcinoma Cells through a Mitogen-Activated Protein Kinases Signal Pathway

**DOI:** 10.3390/ijms232314644

**Published:** 2022-11-24

**Authors:** Yuan Li, Li Yang, Rui Tao, Yajing Shang, Minqiong Sun, Shichao Peng, Guoping Zhao, Ye Zhao

**Affiliations:** 1Teaching & Research Section of Nuclear Medicine, School of Basic Medical Sciences, Anhui Medical University, 81 Meishan Road, Hefei 230032, China; 2Key Laboratory of High Magnetic Field and Ion Beam Physical Biology, Hefei Institutes of Physical Sciences, Chinese Academy of Sciences, 350 Shushanhu Road, Hefei 230031, China

**Keywords:** connexin 26, hepatocellular carcinoma, radiosensitivity, MAPK, NF-κB

## Abstract

Connexin 26 (Cx26) is a protein that constitutes a gap junction and is widely expressed in the liver. Abnormal expression of Cx26 is one of the important mechanisms of liver cancer, and is closely related to the transmission of radiation damage signals between cells. In the present study, we investigated the radiosensitivity of hepatocellular carcinoma (HCC) cells HepG2, with low expression of Cx26, and SK-hep-1, with high expression of Cx26 after X-ray irradiation. The cell survival, micronucleus formation and protein expressions of the mitogen-activated protein kinases (MAPK) signaling pathway were detected. The expression level of Cx26 could affect the radiosensitivity of liver cancer cells by affecting the phosphorylation of p38 and ERK proteins and regulating the expression of downstream NF-κB. Cell lines with knock-out and overexpression of Cx26 were also built to confirm the findings. Our results suggested that Cx26 might play an important role in the radiosensitivity of liver cancer and could be a potential target for clinical radiotherapy of liver cancer.

## 1. Introduction

Hepatocellular carcinoma (HCC) is one of the most common malignant tumors in clinical practice, ranking sixth among the malignant tumors. Additionally, the number of deaths caused by HCC ranks the third among tumor-related deaths [1,2]. The incidence of hepatocellular carcinoma is also very high in China. Liver cancer ranks sixth in cancer incidence and second in tumor-related mortality worldwide, with over half of the new cases and deaths occurring in China [3]. Surgical resection has been considered the first choice for the treatment of liver cancer. However, the high rate of recurrence and rate of metastasis after surgery decrease the survival of the patients [4]. Therefore, non-surgical treatment, including radiotherapy, has attracted more and more attention.

It was believed that normal liver tissue is more sensitive to radiation, and its sensitivity to radiation is only inferior to bone marrow, lymphoid tissue and kidney [5]. In recent years, due to the application of three-dimensional conformal radiotherapy (3D-CRT), stereotactic body radiotherapy (SBRT), hypo-fractionated radiotherapy (HFRT), proton and heavy ion radiotherapy and other technologies in the treatment of HCC, radiotherapy has become more important in the treatment of HCC [6,7,8]. Therefore, a better understanding of the radiosensitivity of HCC cells is of great significance for improving the therapeutic effect of radiotherapy and expanding the application of radiotherapy in HCC treatment.

Connexin 26 (Cx26) is a connexin protein with a molecular weight of 26 kD, which is widely distributed in the human body. Cx26 is a major connexin in mammalian liver, and loss of Cx26 may be closely related to liver carcinogenesis [9,10,11]. Introduction of exogenous Cx26 into liver cancer tissue can induce differentiation of cancer cells [12]. However, Cx26 is also highly expressed in some liver cancer cells, and its expression is closely related to epithelial mesenchymal transition (EMT) of human highly invasive liver cancer cells. Inhibiting the expression of Cx26 can significantly reduce the invasive ability of human highly invasive liver cancer cells in vitro [13].

Radiation-induced damage signals are mediated in large part by gap junction-mediated intercellular communication (GJIC), which can be formed by Cx26 and other connexins [14]. In a previous study, we found that induced expression of Cx26 in HeLa human cervical cancer cells (HeLa^Cx26+/+^) significantly enhanced radiation-induced damage in bystander normal cells by modulating mitogen-activated protein kinases (MAPK) signaling pathway [15]. Additionally, we found that the expression level of Cx26 affects the radiosensitivity of cutaneous squamous cell carcinoma (cSCC) cells [16]. However, there are few data about the role of Cx26 on the radiosensitivity of HCC cells.

In the present study, HepG2 with low expression of Cx26 and SK-hep-1 with high expression of Cx26 were investigated to find the role of Cx26 on the radiosensitivity of HCC cells. These results might improve our understanding of the radiation-induced damage signal pathway and uncover the mechanism underlying the radiosensitivity of HCC cells.

## 2. Results

### 2.1. Expression of Cx26 Differs in Liver Cancer Samples

The expression of Cx26 in liver cancer samples and normal liver samples were analyzed through the Oncomine^TM^ database (Compendia Bioscience, Ann Arbor, MI, USA). The data from five different datasets showed that the expression of Cx26 is lower in liver cancer samples than normal liver samples with substantial differences being observed in at least two datasets (Figure 1, *p* < 0.05). However, some liver cancer samples showed higher Cx26 expression level than normal samples (Figure 1D,E). The survival data of the Guichard dataset, including both high and low expression levels of Cx26, showed liver cancer with higher Cx26 expression had better survival rate than that with lower Cx26 expression (Figure 1F).

### 2.2. Expression of Cx26 in HepG2 and SK-hep-1 Cells

The expression of Cx26 in hepatocellular carcinoma cell HepG2 and SK-hep-1 were detected 6 h after X-ray irradiation by Western blotting. The results showed that the expression of Cx26 in SK-hep-1 were much higher than HepG2 cells under the same condition (Figure 2, *p* < 0.01). However, the expression of Cx26 showed no significant difference after X-ray irradiation between irradiated cells and control cells in neither HepG2 or SK-hep-1.

### 2.3. The Radiosensitivity of HepG2 and SK-hep-1 Cells

HepG2 and SK-hep-1 cells were irradiated with X-rays at doses of 1, 2, 3 and 5 Gy. The clone formation and micronuclei formation assays were used to detect the radiosensitivity of the cells 6 h after irradiation. The dose-survival curves of the clone formation experiments were fitted through a multi-target model. The results showed that the survival fraction of the HepG2 cells (D_0 HepG2_ = 3.35501) was higher than that of the SK-hep-1 cells (D_0 SK-hep-1_ = 2.85814) after X-ray irradiation (Figure 3A). The micronuclei formation rate of cells after radiation was detected by the cytokinesis block method. The micronuclei formation rates were increased in both HepG2 and SK-hep-1cells as the dose increased. However, the micronuclei formation rates of HepG2 cells were much lower than those of SK-hep-1 cells at high doses (*p* < 0.01, Figure 3B). These results suggested that HepG2 cells with low Cx26 expression have lower radiosensitivity than SK-hep-1 cells with high Cx26 expression.

### 2.4. The Activation of the MAPK and NF-κB Signaling Pathways in HepG2 and SK-hep-1 Cells

HepG2 and SK-hep-1 cells were irradiated with X-rays at doses of 1, 2, 3 and 5 Gy. The cells were harvested 6 h after irradiation and the proteins were collected. The expressions of MAPK signaling pathway and downstream NF-κB were detected by Western blotting. The results showed that the expressions of p38 and ERK were not significantly changed in irradiated HepG2 and SK-hep-1 cells compared to sham-irradiated control. However, the phosphorylated p38 (p-p38) and phosphorylated ERK (p-ERK) were increased as the radiation dose increased in both HepG2 and SK-hep-1 cells (Figure 4A,C). The ratios of p-p38/p38 and p-ERK/ERK in SK-hep-1 cells were significantly higher than those in HepG2 cells after 2, 3 and 5 Gy X-ray radiation (Figure 4B,D, *p* < 0.01). The expressions of NF-κB were increased as the radiation dose increasing in both HepG2 and SK-hep-1 cells. Moreover, the expressions of NF-κB were significantly increased in SK-hep-1 cells than those in HepG2 cells after 2, 3 and 5 Gy X-ray radiation (Figure 4E,F, *p* < 0.01). These results suggested that the activation of MAPK and NF-κB signaling pathways were different in HepG2 and SK-hep-1 cells, which might result in the different radiosensitivity of two cells.

### 2.5. Expression of Cx26 Affects the Radiosensitivity of HepG2 and SK-hep-1 Cells

To confirm that the difference in expression of Cx26 affects the radiosensitivity of HepG2 and SK-hep-1 cells, cell lines with knock-out (HepG2^Cx26-^ and SK-hep-1^Cx26-^) and overexpression of Cx26 (HepG2^Cx26+^ and SK-hep-1^Cx26+^) were built (Figure 5A,B). The clone formation and micronuclei formation assays were performed 6 h after irradiation in all cells. The results showed that the survival fractions were increased after knock-out of Cx26 in both cells (D_0 HepG2_ = 3.3550 vs. D_0 HepG2 Cx26-_ = 3.6663; D_0 SK-hep-1_ = 2.8581 vs. D_0 SK-hep-1 Cx26-_ = 3.3550, Figure 5C), while the survival fractions were decreased after overexpression of Cx26 in both cells (D_0 HepG2_ = 3.3550 vs. D_0 HepG2 Cx26+_ = 2.7789; D_0 SK-hep-1_ = 2.8581 vs. D_0 SK-hep-1 Cx26+_ = 2.3878, Figure 5D). The micronuclei formation rates were decreased significantly in both HepG2^Cx26−^ and SK-hep-1^Cx26−^ cells than those of HepG2 and SK-hep-1 cells at high doses (*p* < 0.01, Figure 5E). Additionally, the micronuclei formation rates were increased significantly in HepG2^Cx26+^ cells compared to those of HepG2 cells at high doses. However, the micronuclei formation rates showed no significant increase in SK-hep-1^Cx26+^ cells compared to SK-hep-1 cells (*p* < 0.01, Figure 5F). These results demonstrated that the expression level of Cx26 affects the radiosensitivity of HepG2 and SK-hep-1 cells.

### 2.6. Expression of Cx26 Affects the Activation of the MAPK and NF-κB Signaling Pathways in HepG2 and SK-hep-1 Cells

To find whether the expression of Cx26 affects the MAPK and NF-κB signaling pathways in HepG2 and SK-hep-1 cells, western blotting were performed 6 h after 0, 3 and 5 Gy X-ray irradiation. The results showed that the ratios of p-p38/p38 and p-ERK/ERK were decreased significantly in both HepG2^Cx26−^ and SK-hep-1^Cx26−^ than those of HepG2 and SK-hep-1 cells after irradiation (*p* < 0.01, Figure 6A,B,E,F). Additionally, the ratios of p-p38/p38 and p-ERK/ERK were increased significantly in HepG2^Cx26+^ cells compared to those of HepG2 cells after high dose irradiation (*p* < 0.01, Figure 6C,D,G,H). Only the p-ERK/ERK were increased significantly in SK-hep-1^Cx26+^ cells compared to SK-hep-1 cells after 5 Gy irradiation. The ratios of p-p38/p38 showed no significant difference in SK-hep-1^Cx26+^ cells compared to SK-hep-1 cells. These results might due to the high background expression level of Cx26 in SK-hep-1 cells. Similarly, the expressions of NF-κB were decreased significantly in both HepG2^Cx26−^ and SK-hep-1^Cx26−^ compared to those of HepG2 and SK-hep-1 cells after irradiation (*p* < 0.01, Figure 6I,J). Additionally, the expressions of NF-κB were increased significantly in both HepG2^Cx26+^ and SK-hep-1^Cx26+^ cells compared to those of HepG2 and SK-hep-1 cells after high dose irradiation (*p* < 0.01, Figure 6K,L).

## 3. Discussion

Gap junction allows the free exchange of small molecules (such as ions and second messengers) between adjacent cells. It plays important roles in cell migration, differentiation, proliferation and organ formation [17]. It is by and large accepted that the absence of GJIC is connected with the tumor phenotype. However, in some cases, GJIC can be reduced, maintained or increased in tumor cells. For instance, the expression of Cx40 and Cx26 is up-regulated in testicular malignant tumors, and the expression of Cx43 can be up-regulated or down-regulated according to the tumor type [18]. The expression of Cx43, Cx40, Cx26 and Cx32 is to maintain small cell lung cancer [19]. Epigenetic changes, specifically DNA hypermethylation of its gene promoter, are to blame for the decreased expression of Cx26 in HCC [20]. In these cases, GJIC could play a role in the efficacy of anti-tumor therapy. The gap junction formed by Cx26 mainly transmits radiation damage signals, and can induce the increase of micronucleus rate of cells in the non-irradiated bystander cells [14]. It was found that HeLa cells transfected with Cx26 plasmid could induce the micronucleus formation in non-irradiated bystander cells more rapidly and strongly, enhancing the radiation damage effect. In addition, the radiation damage effect mediated by Cx26 is far greater than that of Cx32, which can directly cause the death of irradiated cells and affected cells in the adjacent area. It induced less genetic damage in offspring cells, and decreased the probability of secondary cancer caused by radiation [15]. In this study, we focused on the changes of radiosensitivity and MAPK signal pathway proteins in HepG2 cells with low expression of Cx26 and SK-hep1 cells with high expression of Cx26. The results showed that HepG2 cells showed a higher survival rate and less micronucleus formation than SK-hep-1 cells after X-ray irradiation. After irradiation, the phosphorylation levels of p38 and ERK1/2 in HepG2 cells were lower than those in SK-hep-1 cells. X-ray irradiation increased micronuclei formation in HepG2^Cx26+^ cells, but no effect was observed in SK-hep-1^Cx26+^ cells. Meanwhile, a significant reduction of micronuclei formation was observed at 5 Gy doses (Figure 5F). In our previous study, we found that Cx26 mainly transmits radiation-induced damage signals. Cells with overexpression of Cx26 could cause more severely damage to neighboring cells after irradiation [15]. SK-hep-1^Cx26+^ cells have high basal level of Cx26 expression. After overexpression of Cx26, the high dose radiation would induce more cell death in SK-hep-1^Cx26+^ cells (Figure 5D). In this case, less micronuclei formation was observed. Additionally, after radiation, the expression of NF-κB in HepG2 cells was lower than that in SK-hep-1 cells. These results indicated that cells with high Cx26 expression show high radiosensitivity. The expression level of Cx26 may affect the phosphorylation of p38 and ERK1/2 and downstream NF-κB expression, regulating the radiosensitivity of HCC cells.

In controlling gene expression, cell survival and cell death, MAPK is an essential signaling pathway [21]. Previous studies show that ERK is required for cell proliferation and differentiation and is primarily activated by growth factors and tumor promoters, whereas p38 promotes cell apoptosis [22,23]. ERK can be triggered by both survival and death signals. ERK plays two roles, depending on the cell type and the stimulus. The duration of ERK activation can be used as a negative regulator of cell survival and a promoter of cell apoptosis in addition to its involvement in cell proliferation and differentiation [24,25]. In addition, MAPK signaling pathway is considered to play an important role in gap junctional communication [26]. Chinese cabbage extract and sulforaphane glucosinolate (SFN) can prevent GJIC inhibition by blocking the inactivation of Cx43 phosphorylase and ERK and p38 MAP kinase [27]. Previous research results have shown that inhibition of GJIC induced by H_2_O_2_ is involved in the activation of ERK and p38 [28,29]. Additionally, Cx32 protein plays a key role in the biological process of hepatocyte proliferation and cell death [30]. It has been demonstrated that oxygen-glucose deprivation, which prevents ERK and p38 MAPK reduction, causes Cx32 up-regulation and damage to hippocampal neurons [31]. Kuan et al. reported that magnolol inhibited ERK-modulated metastatic potential of HCC cells [32]. Weng et al. reported regorafenib slows the progression of hepatocellular carcinoma in mice by inhibiting ERK/NF-κB activation [33]. Our results showed that after Cx26 was up-regulated, the radiosensitivities of both HCC cells were significantly improved. The phosphorylation levels of p38 and ERK and the expression level of NF-κB also increased significantly.

In summary, this study found that after Cx26 was up-regulated, the radiosensitivity of both HepG2 and SK-hep-1 were significantly improved. The phosphorylation levels of ERK1/2 and the expression level of NF-κB increased significantly in both cells. The phosphorylation levels of p38 were increased in HepG2 cells, while not changed significantly in SK-hep-1 cells. On the contrary, after knock-out of Cx26, the radiosensitivity of the two HCC cells were significantly reduced. Additionally, the phosphorylation levels of p38, ERK1/2 and the expression level of NF-κB also decreased significantly. These results further proved that the change of Cx26 expression level may affect the phosphorylation level of key proteins in MAPK signaling pathway, thereby affecting downstream expression of NF-κB that causes the change of cell radiosensitivity. The results revealed the molecular mechanism of the effect of Cx26 on radiosensitivity of HCC cells. Based on this result, the expression level of Cx26 in liver cancer cells of different patients should be detected before radiotherapy of liver cancer, which would be helpful to improve the formulation of radiotherapy plan and the optimization of three-dimensional conformability. According to the different expression levels of Cx26, the radiation dose and radiation area can be adjusted in a targeted way to reduce the damage of radiation to normal liver tissue, and improve the therapeutic effect of radiotherapy on liver cancer while enhancing the killing ability of radiation to liver cancer cells.

## 4. Materials and Methods

### 4.1. Data Acquisition from Oncomine Database

The expression of Cx26 in liver cancer cells samples were acquired from Oncomine platform (http://www.oncomine.org, accessed on 15 September 2019). In this study, Wurmbach Liver, Chen Liver, Guichard Liver, Taga Liver and Guichard Liver2 were selected from Oncomine and used for analysis. The data were processed with GraphPad Prism (vesion8.0.2, GraphPad Software, San Diego, CA, USA). All data were analyzed blindly.

### 4.2. Cell Culture

The HepG2 and SK-hep-1 cells were gifted from Cell Bank, Chinese Academy of Sciences, Shanghai, China. The cells were cultured in Dulbeco’s modified Eagle medium (Dulbeco’s MEM, Gibco, Grand Island, NY, USA) with 10% fetal bovine serum (FBS, HyClone, Logan, UT, USA) plus antibiotic (100 μg/mL streptomycin and 100 U/mL penicillin (Gibco)). The cells were cultured in a humidified 5% CO_2_ incubator at 37 °C (Sanyo, Gunma, Japan).

### 4.3. Irradiation

Irradiation was administered using a 6 MV X-ray Varian 23EX linear accelerator (Varian Inc., Palo Alto, CA, USA) at a dose rate of 5 Gy/min. In total, 8 × 10^5^ cells were seeded into 60 mm dishes for 72 h before irradiation. The medium was changed 30 min before irradiation. After irradiation, the cells were put into a 37 °C incubator for 6 h before the experiments.

### 4.4. Clonogenic Assay

The clonogenic assay was used to determine a cell’s radiosensitivity. Then, 6 h after irradiation, the cells were trypsinized, counted and cultivated into 60 mm cell culture dishes. The numbers of cells seeded in each dish were 300 for 0 and 1 Gy, 500 for 2 and 3 Gy and 1000 for 5 Gy, respectively. For the formation of colonies, the cells were allowed to grow continuously for 10 days. For the purpose of counting, cell colonies were fixed with cell fixation solution (absolute methanol) for 20 min, and stained with Giemsa (Sigma-Aldrich, St. Louis, MO, USA). The clonogenic survival fraction = number of colonies/(the number of seeded × plating efficiency). The ratio of the number of colonies to the number of seeded cells in the untreated control was used as plating efficiency. Survival curves were generated using OriginPro 2015 (version b9.2.257, OriginLab Corporation, Northampton, MA, USA) and fitted with the multiple target single-hit model using the equation: SF = 1 − 1 (1 − e^−D/D0^)^N^, where SF is survival fraction and D is the dose. D_0_ is the average lethal dose, which represents the radiosensitivity of the cell population, i.e., the dose of radiation required for the remaining 37% of cells after irradiation, and N represents the number of the targets in the cell.

### 4.5. Micronuclei Assay

The cytokinesis block method was used to measure the frequency of micronuclei after irradiation. The sham-irradiated and the irradiated cells were trypsinized separately, 6 h after irradiation. In each 35 mm culture dish, approximately 5 × 10^4^ cells were seeded. After 4–6 h, the cultures were incubated with cytochalasin B (Sigma, St. Louis, MO, USA) at a final concentration of 2.5 μg/mL at 37 °C for two cell doubling time. The cells were fixed with fixing solution (methanol:acetic acid = 9:1) for 20 min, then stained with 0.1% acridine orange for 5 min. The cells were examined with a fluorescence microscope (Olympus IX71, Tokyo, Japan). The number of binucleated cells with micronuclei was scored and the frequency of micronuclei cells per 1000 binucleated cells was calculated (% micronuclei).

### 4.6. Western Blot Analysis

The Western blot analyses were taken in direct irradiated cells and the sham-irradiated samples. In the presence of protease and phosphatase inhibitors, total cell lysates were prepared. Using polyacrylamide gel electrophoresis, an equal amount of protein from each sample was resolved and electro-blotted onto PVDF membranes. The monoclonal antibodies used for Western blotting included: anti-ERK1/2; anti-phospho-ERK1/2; anti-p38; anti-phospho-p38; anti-NF-κB; anti-β-actin (Cell Signaling, Beverly, MA, USA); anti-connexin 26 (Santa Cruz, Santa Cruz, CA, USA). The secondary antibodies were conjugated to horseradish peroxidase (Cell Signaling, Beverly, MA, USA); signals were detected using the ECL system (Thermo Scientific, Rockford, IL, USA).

### 4.7. Construction of Cx26 Knock-Out and Overexpression Cells

Cx26 knock-out cells were constructed by transfecting Cx26 CRISPR/Cas9 plasmids (Santa Cruz Biotechnology, Dallas, TX, USA). The gene sequences it knocked out were A: TCGCATTATGATCCTCGTTG and B: GAGCCAGATCTTTCCAATGC. The lyophilized Cx26 plasmid DNA was centrifuged at 12000 RPM for 5 min. Then, 200 μL of ultra-pure sterile DNA-enzyme free water was added to fully dissolve the DNA at a concentration of 0.1 μg/μL. The cells were trypsinized, collected and resuspended in EP Buffer (Opti-MEM medium, Gibco, Grand Island, NY, USA) free of antibiotics and serum. Approximately 2 × 10^5^ cells were mixed with plasmid DNA and thoroughly mixed. After electroporation, monoclones were selected, and the expression of Cx26 was checked. The monoclones with the best knockout results were used for subsequent experiments. For Cx26 overexpression cells, the cells were transfected with lentiviral activation particles (Santa Cruz Biotechnology) according to the manufacture’s protocol.

### 4.8. Statistics

Statistical analysis was performed on the means of the data obtained from at least three independent experiments. The data were presented as means and standard derivations. The significance levels were assessed using Student’s *t*-test. A *p*-value of 0.05 or less between groups was considered statistically significant.

## Figures and Tables

**Figure 1 ijms-23-14644-f001:**
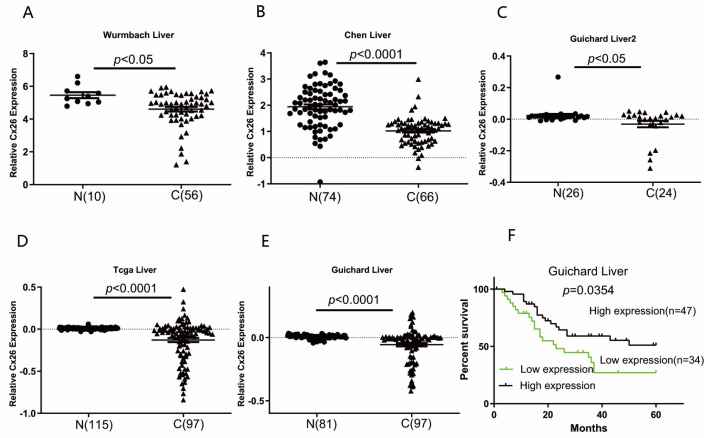
(**A**–**E**) The Cx26 expression levels in liver cancer tissues and normal liver tissue samples were compared using the Oncomine^TM^ database (http://www.oncomine.org/, accessed on 15 September 2019), with 340 samples of liver cancer tissue and 306 normal liver tissue samples. N represents normal liver tissue samples; C represents liver cancer tissue samples. (**F**) The survival data of Guichard dataset.

**Figure 2 ijms-23-14644-f002:**
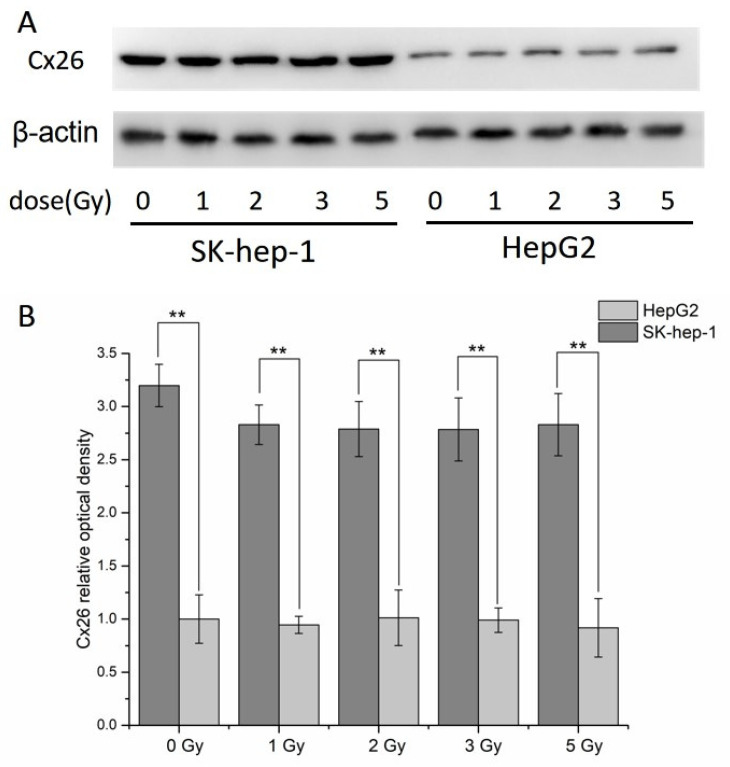
Cx26 expression levels in HepG2 and SK-hep-1 cells 6 h after X-ray irradiation. (**A**) The typical image of Cx26 expression in HepG2 and SK-hep-1 cells 6 h after X-ray irradiation. (**B**) The relative optical density analysis result of Cx26 in HepG2 and SK-hep-1 cells 6 h after X-ray irradiation. ** *p* < 0.01 compared between the two cells.

**Figure 3 ijms-23-14644-f003:**
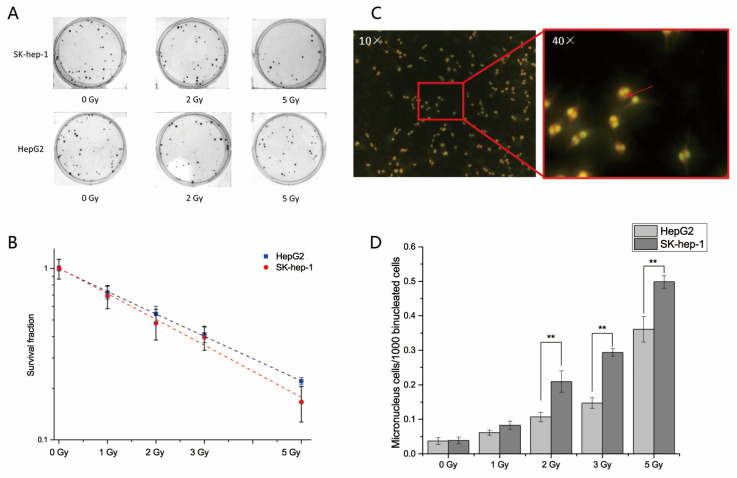
Statistical analysis and schematic diagrams of clone formation and micronuclei formation assays in irradiated HepG2 and SK-hep-1 cells. (**A**) Pictures of stained colonies from the clonogenic assay experiment. (**B**) The dose-survival curve of HepG2 and SK-hep-1 cells 6 h after X-ray irradiation. (**C**) Schematic diagram showing micronuclei formation. (**D**) The micronuclei formation rates of HepG2 and SK-hep-1 cells 6 h after X-ray irradiation. ** *p* < 0.01 compared between the two cells.

**Figure 4 ijms-23-14644-f004:**
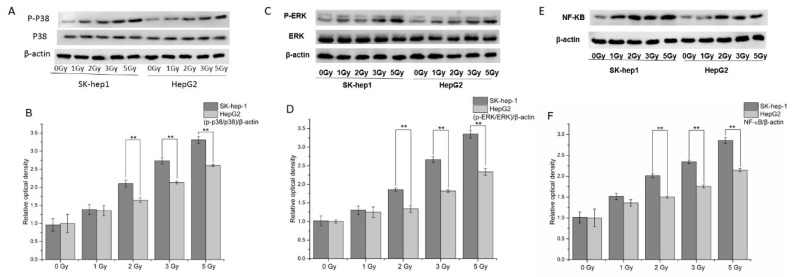
The expression of the MAPK and NF-κB signaling pathways proteins in HepG2 and SK-hep-1 cells after X-ray irradiation. (**A**) The expression of p-p38 and p38 in HepG2 and SK-hep-1 cells 6 h after X-ray irradiation. (**B**) Quantification of p-p38/p38 expression in HepG2 and SK-hep-1 cells 6 h after X-ray irradiation. (**C**) The expression of p-ERK and ERK in HepG2 and SK-hep-1 cells 6 h after X-ray irradiation. (**D**) Quantification of p-ERK/ERK expression in HepG2 and SK-hep-1 cells 6 h after X-ray irradiation. (**E**) The expression of NF-κB in HepG2 and SK-hep-1 cells 6 h after X-ray irradiation. (**F**) Quantification of NF-κB expression in HepG2 and SK-hep-1 cells 6 h after X-ray irradiation. ** *p* < 0.01 compared between the two cells.

**Figure 5 ijms-23-14644-f005:**
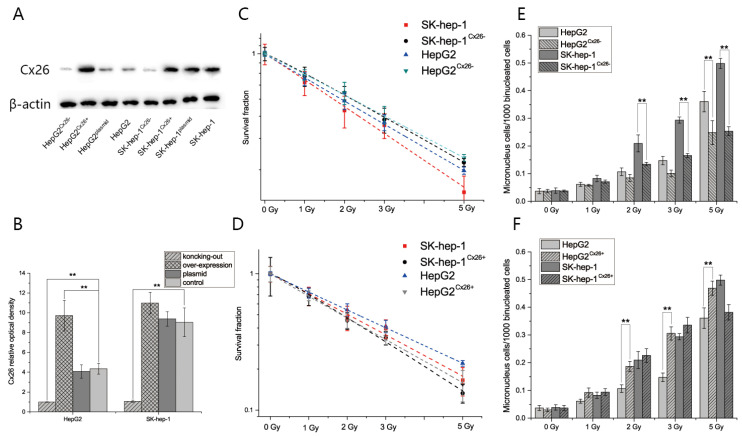
The difference in expression of Cx26 affects the radiosensitivity of HepG2 and SK-hep-1 cells. (**A**) The Cx26 expression in HepG2 and SK-hep-1 cells after knock-out or overexpression of Cx26. (**B**) Quantification of Cx26 expression in HepG2 and SK-hep-1 cells after knock-out or overexpression of Cx26. (**C**) The dose-survival curve of HepG2^Cx26−^ and SK-hep-1^Cx26−^ cells compared with control cells after X-ray irradiation. (**D**) The dose-survival curve of HepG2^Cx26+^ and SK-hep-1^Cx26+^ cells compared with control cells after X-ray irradiation. (**E**) The micronuclei formation rates of HepG2^Cx26−^ and SK-hep-1^Cx26−^ cells after X-ray irradiation. (**F**) The micronuclei formation rates of HepG2^Cx26+^ and SK-hep-1^Cx26+^ cells after X-ray irradiation. ** *p* < 0.01 compared between the two cells.

**Figure 6 ijms-23-14644-f006:**
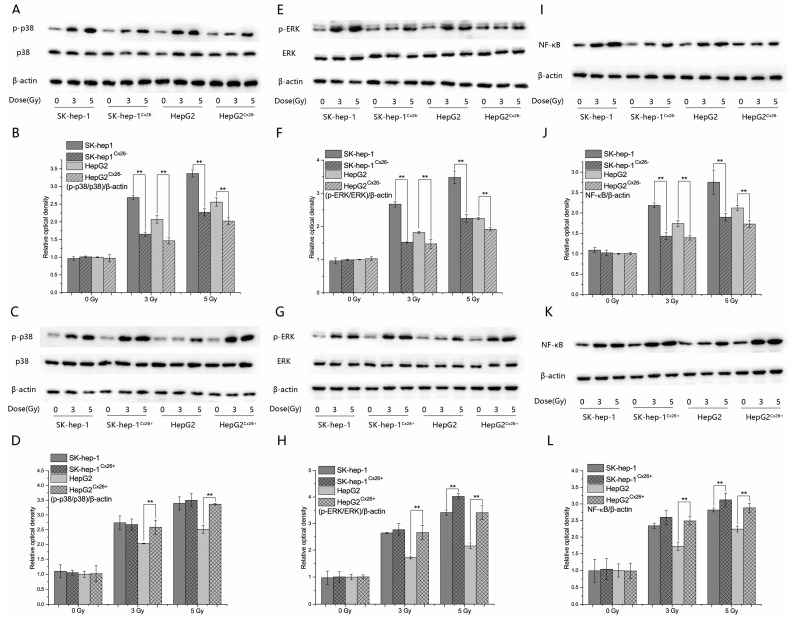
The expression of the MAPK and NF-κB signaling pathways proteins in HepG2, HepG2^Cx26−^, HepG2^Cx26+^, SK-hep-1, SK-hep-1^Cx26−^ and SK-hep-1^Cx26+^ cells after X-ray irradiation. (**A**) The expression of p-p38 and p38 in HepG2^Cx26−^ and SK-hep-1^Cx26−^ cells after X-ray irradiation. (**B**) Quantification of p-p38/p38 expression in HepG2^Cx26−^ and SK-hep-1^Cx26−^ cells after X-ray irradiation. (**C**) The expression of p-p38 and p38 in HepG2^Cx26+^ and SK-hep-1^Cx26+^ cells after X-ray irradiation. (**D**) Quantification of p-p38/p38 expression in HepG2^Cx26+^ and SK-hep-1^Cx26+^ cells after X-ray irradiation. (**E**) The expression of ERK/ERK in HepG2^Cx26−^ and SK-hep-1^Cx26-^ cells after X-ray irradiation. (**F**) Quantification of ERK/ERK expression in HepG2^Cx26−^ and SK-hep-1^Cx26−^ cells after X-ray irradiation. (**G**) The expression of ERK/ERK in HepG2^Cx26+^ and SK-hep-1^Cx26+^ cells after X-ray irradiation. (**H**) Quantification of ERK/ERK expression in HepG2^Cx26+^ and SK-hep-1^Cx26+^ cells after X-ray irradiation. (**I**) The expression of NF-κB in HepG2^Cx26−^ and SK-hep-1^Cx26−^ cells after X-ray irradiation. (**J**) Quantification of NF-κB expression in HepG2^Cx26−^ and SK-hep-1^Cx26−^ cells after X-ray irradiation. (**K**) The expression of NF-κB in HepG2^Cx26+^ and SK-hep-1^Cx26+^ cells after X-ray irradiation. (**L**) Quantification of NF-κB expression in HepG2^Cx26+^ and SK-hep-1^Cx26+^ cells after X-ray irradiation. ** *p* < 0.01 compared between the two cells.

## Data Availability

Data and materials will be made available upon reasonable request.

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
