# Peer review of "The Expression of Connexin 26 Regulates the Radiosensitivity of Hepatocellular Carcinoma Cells through a Mitogen-Activated Protein Kinases Signal Pathway"

_ijms, 2022, doi:10.3390/ijms232314644_

Round 1

Reviewer 1 Report

The original article titled " The Expression of Connexin26 Regulates the Radiosensitivity 2 of Hepatocellular Carcinoma Cells through Mitogen-Activated 3 Protein Kinases Signal Pathway'' was written very well, suggests Cx26 might play an important role in the radio- 25 sensitivity of liver cancer and could be a potential target for clinical radiotherapy of liver cancer.

Author Response

Answer: Thank you very much for your comments.

Reviewer 2 Report

Hepatocellular carcinoma (HCC) is the second most common cause of cancer death worldwide, with a majority of HCC patients not suitable for curative therapies as surgery resection. Radiotherapy can offer high local control rates in unresectable HCC and palliation in metastatic cases. In this report, Li and colleagues investigate the role of gap junctions on the radiosensitivity of HepG2 and SK-hep-1 cell lines. The authors conclude that the upregulation of the gap junction beta-2 protein (connexin 26, Cx26), promotes the activation of key proteins in the MAPK signaling pathway (p-38 and ERK1/2), increased expression of is downstream target NF-κB, and improves the radiosensitivity of both cells lines, Opposite results were observed in Cx26 knockout.

Major issues:

Some conclusions are not supported by the results (Lines 246-247). Overexpression of Cx26 in SK-hep-1 cells does not affect p38 activation after X-ray irradiation (Figure 6D).

X-ray irradiation increased micronuclei formation in HepG2Cx26+ cells but no effect was observed in SK-hep-1Cx26+ cells, however, a significant reduction is observed at 5 Gy doses (Figure 5F, SK-hep-1Cx26+). Discuss.

The use of HepG2 cells as an HCC model is incorrect (see Arzumanian et al. 2021, PMID: 34884942). Alternative HCC cell lines must be used.

Considering the suggested potential use of X-ray radiation doses adjusted according to different expression levels of Cx26 as a strategy for HCC-treatment, Cx26 knockout and overexpression experiments must be validated in vivo and/or primary hepatocytes in control vs. HCC-induced models.

Minor:

Line 70; Oncomine TM database, transfer to material and methods and include a brief description of the analysis

Lines 94, 112; …at doses of 0, 1, 2, 3 and 5 Gy. => …at doses of 1, 2, 3 and 5 Gy.

Line 97; multi-target click model?, include a brief description (D0) in material and methods

Line 196; Gap junction is an important channel of intercellular communication, which… => Gap junctions, which…

Lines 205. 206; …In these cases, people will expect GJIC to play… => … In these cases, GJIC could play…

Line 214: experiment => study

Line 225: Other studies… => Previous studies…

Line 233; define SFN

Line 271; …2×105 cells/mL, and the cells were seeded into 60-mm... => …2x105 cells seeded into 60 mm…

Line 275; include the number of cells seeded

Line 287; Φ35 mm => 35 mm

Lines 292, 293; …of micronuclei cells in binucleate… => …the percentage of binucleated cells containing micronuclei (% micronuclei)…

Figure 3. Include a panel with representative images of binucleated HepG2 and SK-hep-1 cells (control and after X-ray irradiation, as evidence of micronuclei formation)

Figures 3B, and 5E, F; Micronucleus cells/1000 binucleated cells => % micronuclei

Figures 4B,D, and 6D,H; p-p38/p38/β-actin, pERK/ERK/β-actin => (p-p38/p38)/β-actin, (pERK/ERK)/β-actin

Avoid the use of unnecessary abbreviations (SBRT), (HFRT), (BN).

Normalize color code in bar graphs; Figure 4, HepG2 (light grey) SK-hep-1 (dark grey)

Typo & text errors

Connexin26 => Connexin 26

Line 72; caner => cancer

Line 97; multi-target click model => multi-target model

Line 149; SK-hep-1Cx26- => SK-hep-1Cx26+

Line 300; anti-phosph-p38 => anti-phospho-p38

Line 310; opTI-MEm => Opti-MEM

Figure 2A; hepG2 => HepG2

Figure 5A; actin => β-actin

Author Response

Hepatocellular carcinoma (HCC) is the second most common cause of cancer death worldwide, with a majority of HCC patients not suitable for curative therapies as surgery resection. Radiotherapy can offer high local control rates in unresectable HCC and palliation in metastatic cases. In this report, Li and colleagues investigate the role of gap junctions on the radiosensitivity of HepG2 and SK-hep-1 cell lines. The authors conclude that the upregulation of the gap junction beta-2 protein (connexin 26, Cx26), promotes the activation of key proteins in the MAPK signaling pathway (p-38 and ERK1/2), increased expression of is downstream target NF-κB, and improves the radiosensitivity of both cells lines, Opposite results were observed in Cx26 knockout.

Major issues:

Some conclusions are not supported by the results (Lines 246-247). Overexpression of Cx26 in SK-hep-1 cells does not affect p38 activation after X-ray irradiation (Figure 6D).

Response: Thank you very much for your suggestions. We have revised the manuscript according to your suggestions.

X-ray irradiation increased micronuclei formation in HepG2Cx26+ cells but no effect was observed in SK-hep-1Cx26+ cells, however, a significant reduction is observed at 5 Gy doses (Figure 5F, SK-hep-1Cx26+). Discuss.

Response: Thank you very much for your question. In our previous study, we found that Cx26 mainly transmits radiation induced damage signals. Cells with over expression of Cx26 could cause more severely damage to neighboring cells after irradiation. SK-hep-1Cx26+ cells have high basal level of Cx26 expression. After overexpression of Cx26, the high dose radiation would induce more cell death in SK-hep-1Cx26+ cells (fig5D). In this case, less micronuclei formation was observed. We have also added these in the discussion section.

.

The use of HepG2 cells as an HCC model is incorrect (see Arzumanian et al. 2021, PMID: 34884942). Alternative HCC cell lines must be used.

Response: Thank you very much for your suggestions. According to Arzumanian et al. (2021, PMID: 34884942), the validity of using HepG2 cells as a hepatocyte model is controversial. Because the main functions of the liver, a key protein involved in the metabolism of substances, are poorly expressed. In addition, the deficiency of uptake transporters and phase I enzymes was observed in HepG2 cells, which indicated the need for the careful use of this cell line to predict the metabolism and elimination of xenobiotics in hepatocytes. But the author admit that these are not important for some work, such as studying the number of proteins in a cell. In present study, X-rays were mainly used as the treatment method to detect the role of Cx26 in radiosensitivity of liver cancer cells, and no drug metabolism-related mode was involved. HepG2 cell expressed low base level of Cx26 which is suitable for our study.

Considering the suggested potential use of X-ray radiation doses adjusted according to different expression levels of Cx26 as a strategy for HCC-treatment, Cx26 knockout and overexpression experiments must be validated in vivo and/or primary hepatocytes in control vs. HCC-induced models.

 Response: Thank you very much for your suggestions. HepG2 and SK-hep-1 cells used throughout this study were human tumor cell lines. For in vivo study, a widely used model is subcutaneous injection of human HCC cell lines in immunodeficient mice, usually athymic nude or severe combined immunodeficient (SCID) mice. However, immunodeficient mice were more radiosensitive because of lacking immune cells, which might cause different results compared to normal animal. Moreover, the radiosensitivity of mice is higher than human being. So, we only use HepG2 and SK-hep-1 cells to study the role of Cx26 on radiosensitivity of liver cancer cells in this study.

Minor:

Line 70; Oncomine TM database, transfer to material and methods and include a brief description of the analysis

Response: Thank you very much for your suggestions. We have revised the manuscript, and transfer this to the material and methods section.

Lines 94, 112; …at doses of 0, 1, 2, 3 and 5 Gy. => …at doses of 1, 2, 3 and 5 Gy.

Response: Thank you very much for your suggestions. We have revised the manuscript according to your suggestions.

Line 97; multi-target click model?, include a brief description (D0) in material and methods

Response: Thank you very much for your suggestions. We have revised the manuscript, and added the description of D0 in material and methods.

Line 196; Gap junction is an important channel of intercellular communication, which… => Gap junctions, which…

Response: Thank you very much for your suggestions. We have revised the manuscript according to your suggestions.

Lines 205. 206; …In these cases, people will expect GJIC to play… => … In these cases, GJIC could play…

Response: Thank you very much for your suggestions. We have revised the manuscript according to your suggestions.

Line 214: experiment => study

Response: Thank you very much for your suggestions. We have revised the manuscript according to your suggestions.

Line 225: Other studies… => Previous studies…

Response: Thank you very much for your suggestions. We have revised the manuscript according to your suggestions.

Line 233; define SFN

Response: Thank you very much for your suggestions. We have revised the manuscript, and added the definition of SFN in our revised manuscript.

Line 271; …2×105 cells/mL, and the cells were seeded into 60-mm... => …2x105 cells seeded into 60 mm…

Response: Thank you very much for your suggestions. We have revised the manuscript according to your suggestions.

Line 275; include the number of cells seeded

Response: Thank you very much for your suggestions. We have revised the manuscript, and added the number of cells seeded in our revised manuscript.

Line 287; Φ35 mm => 35 mm

Response: Thank you very much for your suggestions. We have revised the manuscript according to your suggestions.

Lines 292, 293; …of micronuclei cells in binucleate… => …the percentage of binucleated cells containing micronuclei (% micronuclei)…

Response: Thank you very much for your suggestions. We have revised the manuscript according to your suggestions.

Figure 3. Include a panel with representative images of binucleated HepG2 and SK-hep-1 cells (control and after X-ray irradiation, as evidence of micronuclei formation)

Response: Thank you very much for your suggestions. The micronuclei are usually small, which could not be seen clearly in low lens (10× or 20×). However, the cell number is very low in one scope field in high lens (40×). At the same time, the micronuclei rate is relatively low that only one or two cells with micronuclei could be seen in one filed. These make it difficult to observe the difference of micronuclei formation from images. So, we added the representative images of binucleated cells in our revised manuscript.

Figures 3B, and 5E, F; Micronucleus cells/1000 binucleated cells => % micronuclei 

Response: Thank you very much for your suggestions. We have revised the figures according to your suggestions.

Figures 4B,D, and 6D,H; p-p38/p38/β-actin, pERK/ERK/β-actin => (p-p38/p38)/β-actin, (pERK/ERK)/β-actin

Response: Thank you very much for your suggestions. We have revised the figures according to your suggestions.

Avoid the use of unnecessary abbreviations (SBRT), (HFRT), (BN).

Response: Thank you very much for your suggestions. We have revised the manuscript according to your suggestions.

Normalize color code in bar graphs; Figure 4, HepG2 (light grey) SK-hep-1 (dark grey)

Response: Thank you very much for your suggestions. We have revised the figures and used respective same color code for HepG2 and SK-hep-1 cells in all figures.

Typo & text errors

Connexin26 => Connexin 26

Line 72; caner => cancer

Line 97; multi-target click model => multi-target model

Line 149; SK-hep-1Cx26- => SK-hep-1Cx26+

Line 300; anti-phosph-p38 => anti-phospho-p38

Line 310; opTI-MEm => Opti-MEM

Figure 2A; hepG2 => HepG2

Figure 5A; actin => β-actin

Response: Thank you very much for your suggestions. We have corrected these errors in our revised manuscript.

Reviewer 3 Report

Li et al. investigate the significance of the gap junction protein Cx26 in sensitizing liver cancer cells to radiation. The study draws merit as it outlines the basic importance of this protein in sensitizing cells to radiation damage. The authors confirm this by assessing the primary stress response MAPK pathway in Cx26 overexpressing and Cx26-depleted cells. Additional experiments are strongly encouraged to characterize the differential expression in cancer vs normal contexts and the role of this protein in promoting radiosensitivity of cancer cells.

-There are minor but visible grammar errors that need to be rectified. Framing of several sentences in the legends and results section need appropriate scientific adaptation.   

-Line 36 states an important statistic but is missing a citation/reference.

-Line 72: correct the typo “caner”. Must write “liver cancer samples”

-Fig C, D and E do not show substantial differences in Cx26 expression in terms of fold-change. The authors should revise text to along the lines of “…with substantial differences being observed in at least two datasets”

-Fig 1F: Include statistical details (p-value)   

-Preliminary mRNA evidence mined from databases are fine but the authors are strongly advised to screen more cell lines and include additional cancer and non-cancer cell lines/primary cells. Alternatively, IHC on tumor microarrays containing normal adjacent controls are advised.

-The authors must show pictures of stained colonies from the clonogenic assay expt.

-Line 141: Knock-out of Cx26 is extremely poor in HepG2 but surprising to see significant impact on radiosensitivity.

-Fig 5E/F: Consider using different contrasting colors for each bar.  

-What is the subcellular distribution of Cx26 in irradiated vs non-irradiated cells? Is it possible that impact the subcellular localization of Cx26 impacts downstream signaling.

-The authors should examine potentially impaired DNA repair signaling in Cx26-depleted cells. Appropriate and convincing evidence includes cell cycle transition and evaluating activation of ATM/ATR.    

-Does modulation of Cx26 expression alone have any impact on cell proliferation and migration of liver cancer cells?  

Author Response

Li et al. investigate the significance of the gap junction protein Cx26 in sensitizing liver cancer cells to radiation. The study draws merit as it outlines the basic importance of this protein in sensitizing cells to radiation damage. The authors confirm this by assessing the primary stress response MAPK pathway in Cx26 overexpressing and Cx26-depleted cells. Additional experiments are strongly encouraged to characterize the differential expression in cancer vs normal contexts and the role of this protein in promoting radiosensitivity of cancer cells.

-There are minor but visible grammar errors that need to be rectified. Framing of several sentences in the legends and results section need appropriate scientific adaptation.   

Response: Thank you very much for your suggestions. We have revised the manuscript according to reviewers’ and editor’s comments.

-Line 36 states an important statistic but is missing a citation/reference.

Response: Thank you very much for your suggestions. We have revised the manuscript, and added the reference in our revised manuscript.

-Line 72: correct the typo “caner”. Must write “liver cancer samples”

Response: Thank you very much for your suggestions. We have revised the manuscript according to your suggestions.

-Fig C, D and E do not show substantial differences in Cx26 expression in terms of fold-change. The authors should revise text to along the lines of “…with substantial differences being observed in at least two datasets”

Response: Thank you very much for your suggestions. We have revised the manuscript according to your suggestions.

-Fig 1F: Include statistical details (p-value)   

Response: Thank you very much for your suggestions. We have revised the figure according to your suggestions.

-Preliminary mRNA evidence mined from databases are fine but the authors are strongly advised to screen more cell lines and include additional cancer and non-cancer cell lines/primary cells. Alternatively, IHC on tumor microarrays containing normal adjacent controls are advised.

Response: Thank you very much for your suggestions. The similar radiosensitivity between liver cancer cells and normal liver cells is the key point that limits the application of radiotherapy in treatment of liver cancer. In this study, we focused on the role of Cx26 on the radiosensitivity of liver cancer cells. The HepG2 cell have low basal level of Cx26 expression while SK-hep-1 cells have high basal level of Cx26 expression. So we choose these two cells to investigate the role of Cx26 on the radiosensitivity of liver cancer cells. Beside liver cancer cells,we have already studied the role of Cx26 on radiosensitivety of skin cancer cells and normal skin cells (Sun et al. 2021, PMID: 34291047). Results from this study provide a preliminary laboratory evidence that Cx26 could modulate the radiosensitivity of liver cancer cells. Further study of clinical patients’ sample or liver cancer case would be the focus of our subsequent studies.

-The authors must show pictures of stained colonies from the clonogenic assay expt.

Response: Thank you very much for your suggestions. We have revised the figures and added the pictures of stained colonies in our revised manuscript.

-Line 141: Knock-out of Cx26 is extremely poor in HepG2 but surprising to see significant impact on radiosensitivity.

Response: Thank you very much for your question. The basal level of Cx26 is relatively low in HepG2 cells. Further knocking out of Cx26 in HepG2 cells is not significant especially seen from image of western blot result. Still, the expression of Cx26 is only about a quarter in HepG2Cx26- cells compared to control HepG2 cells (Fig 5B). The gap junctions composed by Cx26 mainly transmit radiation induced damage signal. Knocking out of Cx26 would resulted in decreasing of cell damage because fewer damage signal between adjacent cells. So, the clone formation is increased while micronuclei formation is decreased in HepG2Cx26- cells after X-ray irradiation, significantly at 5 Gy.

-Fig 5E/F: Consider using different contrasting colors for each bar.  

Response: Thank you very much for your suggestions. We have revised the figures and used respective same color code for HepG2 and SK-hep-1 cells in all figures.

-What is the subcellular distribution of Cx26 in irradiated vs non-irradiated cells? Is it possible that impact the subcellular localization of Cx26 impacts downstream signaling.

Response: Thank you very much for your question. The core concern of our study is whether the different expression of Cx26 have an impact on radiosensitivity of liver cancer cells. Meanwhile, Cx26-composed gap junctional intercellular communication enhances the intercellular propagation of “death signal”, thereby increasing radiosensitivity (Autsavapromporn et al. 2013, PMID: 23867854). The core of our research is the influence of radiation induced damage signal transmission among the whole cell population. At the same time, we also noticed that there are few studies on the subcellular localization of Cx26 after radiation. But there are obvious differences in the expression amount and plasma membrane distribution of Cx26 in different stages of cancer, which may be the focus of our subsequent studies (Ezumi et al. 2008, PMID: 18245526).

-The authors should examine potentially impaired DNA repair signaling in Cx26-depleted cells. Appropriate and convincing evidence includes cell cycle transition and evaluating activation of ATM/ATR.    

Response: Thank you very much for your question. DNA damage repair involves subsequent tumor prognosis. This study is mainly to investigate the impact of Cx26 expression on radiation damage in liver cancer cells. Increases in clonogenic survival and decreased rate of micronucleus were observed in Cx26 low-expressing cancer cells in this study. DNA damage repair may provide a good direction for our future radiotherapy for targeting Cx26 low-expressing tumors.

-Does modulation of Cx26 expression alone have any impact on cell proliferation and migration of liver cancer cells?  

Response: Thank you very much for your question. Connexins are generally considered tumor-suppressive. However, previous studies of clinical samples suggested a different role of connexins in that expression levels and membrane localization of connexins, including Cx43(GJA1) and Cx26 (GJB2), were found to be enhanced in metastatic lesions of cancer patients. Cx43- and Cx26-mediated GJIC was found to promote cancer cell migration and adhesion to the pulmonary endothelium (Wu et al. 2019, PMID: 30642339). Previous studies suggest that loss of Cx26 predisposes the mammary gland to chemically induced mammary tumour formation which may have important implications to patients with GJB2 mutations (Stewart et al. 2015, PMID: 26439696).

Round 2

Reviewer 2 Report

The authors have reasonably addressed the issues raised in my previous review.

Reviewer 3 Report

Comments fairly justified but the authors are encouraged to see that future findings are better characterized.